# TRPV4 and KRAS and FGFR1 gain-of-function mutations drive giant cell lesions of the jaw

Carolina Cavalieri Gomes [1,2], Tenzin Gayden [1], Andrea Bajic [1], Osama F. Harraz [3], Jonathan Pratt[1], Hamid Nikbakht [1,4], Eric Bareke [1,4], Marina Gonçalves Diniz [5], Wagner Henriques Castro[5], Pascal St-Onge[6], Daniel Sinnett [6,7], HyeRim Han[8], Barbara Rivera[8,9], Leonie G. Mikael[10], Nicolas De Jay[1], Claudia L. Kleinman[1,8], Elvis Terci Valera [1,11], Angelia V. Bassenden[12], Albert M. Berghuis, Jacek Majewski [1,4], Mark T. Nelson[3,13], Ricardo Santiago Gomez [1,5] & Nada Jabado [1,10]

Giant cell lesions of the jaw (GCLJ) are debilitating tumors of unknown origin with limited available therapies. Here, we analyze 58 sporadic samples using next generation or targeted sequencing and report somatic, heterozygous, gain-of-function mutations in *KRAS, FGFR1*, and p.M713V/I-*TRPV4* in 72% (42/58) of GCLJ. *TRPV4* p.M713V/I mutations are exclusive to central GCLJ and occur at a critical position adjacent to the cation permeable pore of the channel. Expression of TRPV4 mutants in HEK293 cells leads to increased cell death, as well as increased constitutive and stimulated channel activity, both of which can be prevented using TRPV4 antagonists. Furthermore, these mutations induce sustained activation of ERK1/2, indicating that their effects converge with that of *KRAS* and *FGFR1* mutations on the activation of the MAPK pathway in GCLJ. Our data extend the spectrum of TRPV4 channelopathies and provide rationale for the use of TRPV4 and RAS/MAPK antagonists at the bedside in GCLJ.

[1] Department of Human Genetics, McGill University, Montreal H3A 0C7 QC, Canada. [2] Department of Pathology, Biological Sciences Institute, Universidade Federal de Minas Gerais, Belo Horizonte 31270901 Minas Gerais, Brazil. [3] Department of Pharmacology, Larner College of Medicine, University of Vermont, Burlington 05405 VT, USA. [4] McGill University and Genome Quebec Innovation Centre, Montreal H3A 0G1 QC, Canada. [5] Department of Oral Surgery and Pathology, Faculty of Dentistry, Universidade Federal de Minas Gerais, Belo Horizonte 31270901 Minas Gerais, Brazil. [6] CHU Sainte-Justine Research Center, Université de Montréal, Montreal H3T 1C5 QC, Canada. [7] Department of Pediatrics, University of Montreal, Montreal H3T 1C5 QC, Canada. [8] Lady Davis Research Institute, Jewish General Hospital, Montreal H3T 1E2 QC, Canada. [9] Gerald Bronfman Department of Oncology, McGill University, Montreal H4A 3T2 QC, Canada. [10] Department of Pediatrics, McGill University and McGill University Heath Centre Research Institute, Montreal H4A 3J1 QC, Canada. [11] Department of Pediatrics, Ribeirão Preto Medical School, Universidade de São Paulo, São Paulo H3G 1Y6, Brazil. [12] Department of Biochemistry, McGill University, Montreal M13 9NT QC, Canada. [13] Institute of Cardiovascular Sciences, University of Manchester, Manchester M13 9PL, UK. These authors contributed equally: Carolina Cavalieri Gomes, Tenzin Gayden, Andrea Bajic. These authors jointly directed this work: Ricardo Santiago Gomez, Nada Jabado. Correspondence and requests for materials should be addressed to R.S.G. (email: rsgomez@ufmg.br) or to N.J. (email: nada.jabado@mcgill.ca)

Giant-cell lesions of the jaw (GCLJ) are benign tumors with an often aggressive and unpredictable clinical course[1]. Initially termed as *central giant cell reparative granuloma* to distinguish them from giant cell tumors of the bone[2] (GCTB), their classification was refined into GCLJ by the World Health Organization based on the destructive nature and recurrent pattern[3]. GCLJ are traditionally divided into central and peripheral forms, and are histologically very similar to GCTB, being one of their osteoclast-rich mimics in the jaw. Central GCLJ is an intramedullary bone lesion that affects mainly the anterior mandible of young patients. The peripheral form occurs in older individuals, predominantly between 40 and 60 years of age, and affects mainly the mandible, with a recurrence rate of approximately 20%[4]. The histopathological features of GCLJ consist of a main tumor component represented by mononuclear spindle-shaped and polygonal cells, in addition to the pathognomonic multinucleated giant cells in a vascular background[5]. Tumors are classified as aggressive or nonaggressive depending on size, growth pattern, tooth resorption or displacement, cortical bone destruction or thinning, and based on recurrence[6–8]. Even if potentially debilitating with serious facial mutilations in some cases, surgical removal is the mainstay of therapy. However, aggressive forms of GCLJ show frequent escape from this traditional surgical management and limited response to adjuvant therapies including corticosteroids. These are painful, rapidly growing and bone perforating recurrent lesions with major functional impact on the jaw and teeth structure[6,9]. Moreover, GCLJ do not have high receptor activator of nuclear-factor κB ligand (RANKL) expression in contrast to the close GCTB[5], making the use of costly targeted inhibitors to this receptor difficult to propose, despite a recent report showing tumor regression in five GCLJ cases[10].

One barrier to alternate and more effective therapeutic strategies is the limited information on molecular drivers of GCLJ. Although they mimic osteoclast-rich GCTBs, these tumors lack the recurrent somatic *H3F3A* mutations described in this entity[11–13]. To uncover pathogenic drivers of the disease, we analyzed 58 GCLJ samples (central form $n = 37$, peripheral form $n = 21$), performed next generation sequencing (NGS) and targeted sequencing on these samples, and further validated the targets we identified using functional assays. Our data show that recurrent, heterozygous, somatic transient receptor potential vanilloid 4 cation channel (*TRPV4*) p.M713V and p.M713I, *KRAS* and *FGFR1* mutations are the most relevant genetic alterations at the basis of GCLJ. These mutations occur in 72% (42/58) of tumors and converge in their effects on activating the MAPK pathway, including the *TRPV4* p.M713V and p.M713I amino acid substitutions, as we show herein.

## Results

**Driver mutations in GCLJ**. We accrued samples from central and peripheral forms of GCLJ (Fig. 1a, Supplementary Data 1) and performed NGS on 19 tumors (whole-exome sequencing (WES) $n = 18$; RNA-Seq $n = 6$; Supplementary Data 1, Supplementary Fig. 1). Tumor mutation burden was low (1 per Mb), as determined for five cases for which WES was performed for tumor and matched normal DNA (Supplementary Data 2). This is consistent with the benign nature of these lesions and matches previous findings on the closely related GCTB[11]. Analysis of the datasets identified nucleotide substitutions in *TRPV4* leading to p. M713V or p.M713I in three samples, two amino acid changes on the same residue. *TRPV4* encodes a broadly expressed polymodal $Ca^{2+}$-permeable channel and germline heterozygous dominant mutations across this gene have been identified in a wide range of diseases, but not in GCLJ or related bone disorders

(Supplementary Fig. 2)[14]. We also identified previously described multiple *KRAS* mutations in nine samples and two *FGFR1* mutations in three additional samples, while four samples were wild-type (WT) for these genes (triple negatives) (Fig. 1b, Supplementary Data 1, Supplementary Fig. 1). To validate these mutations, we performed targeted sequencing using Sanger sequencing and, whenever possible, MiSeq analysis on these and 39 additional GCLJ samples (Fig. 1b, Supplementary Data 1, Supplementary Fig. 1). Sequencing results showed that recurrent, heterozygous, mutations in *TRPV4*, *KRAS*, and *FGFR1* occur in 72.4% (42/58) GCLJ (Fig. 1b, c, Supplementary Figs. 2 and 3, Supplementary Data 1). These mutations were somatic in all patients with germline material available and showed variable reads ranging from 10 to 64% in samples analyzed using deep sequencing (Supplementary Data 1). The low-mutational read observed in a few samples also mirrors findings in the close-related GCTB. Indeed, in this entity the driver *H3F3A* mutation, which is only present in the stromal and not in giant cells component of the tumor, shows similar low reads in a subset of tumors[11]. Sixteen samples in our cohort were WT for *TRPV4*, *KRAS*, and *FGFR1*. As Sanger sequencing can typically detect mutations present in ~20% of cells in a given sample, this triple-negative status was confirmed using NGS in five tumors with available material (WES ($n = 3$) and/or MiSeq ($n = 5$)). In the remaining 11 cases assessed by Sanger sequencing only, we cannot exclude false negatives based on possible low mutation reads or sampling issues (Fig. 1b, c, Supplementary Data 1, Supplementary Fig. 1). We did not identify other recurrent genetic alterations in these WT samples or in samples carrying *TRPV4*, *KRAS*, or *FGFR1* mutations (Supplementary Datas 3–5).

**TRPV4 mutations lead to increased channel activity**. Somatic *TRPV4* mutations were identified in 22% (13/58) of tumors, exclusively in the central form of GCLJ, and led to p.M713V ($n = 11$) and p.M713I ($n = 2$) (Fig. 1, Fig. 2a, Supplementary Data 1). Neither mutation has ever been reported in germline TRPV4 channelopathies[14] or in other diseases, including cancer, except for one renal cell carcinoma case where the M713I mutation was listed without functional characterization among numerous other genetic alterations[15] (Supplementary Fig. 2, Supplementary Data 6). There was no association between *TRPV4* mutation status and clinical aggressiveness, or tumor location in mandible or maxilla (Fig. 1b). We also detected rare *TRPV4* variants p. A431T (rs955455114; $n = 3$) and p.Y283N (rs200210023; $n = 1$), but based on their relative frequency in the general population, the lack of potential functional impact by in silico modeling and their presence in the germline in one individual with no other clinical manifestations, these were not pursued further.

Based on their high frequency in GCLJ and the absence of known functional data, we further investigated M713V/I-TRPV4 mutations. M713 is located at a critical position in the sixth transmembrane domain adjacent to the TRPV4 channel pore (Fig. 2a). To predict whether p.M713V and p.M713I mutations affect channel function, we performed in silico modeling using the published structure of the closely related TRPV1[16,17] (Fig. 2b). Based on the open and closed state models of TRPV4, residue M713 is located at the interfaces between the four monomers of the TRPV4 homo-tetramer, on helix S6, in a hydrophobic cleft adjacent to helix S5. The residues in helices S5 and S6 are conserved between TRPV1 and TRPV4. Helix S6 slides with respect to helix S5 during channel opening and closing. Residue M677 of TRPV1, which corresponds to M713 in TRPV4, was altered in both open and closed states of the channel[16,17] (Fig. 2c, d). This implies that TRPV4 mutations at p.M713 may affect the relative stabilities of open and closed states of the channel and the

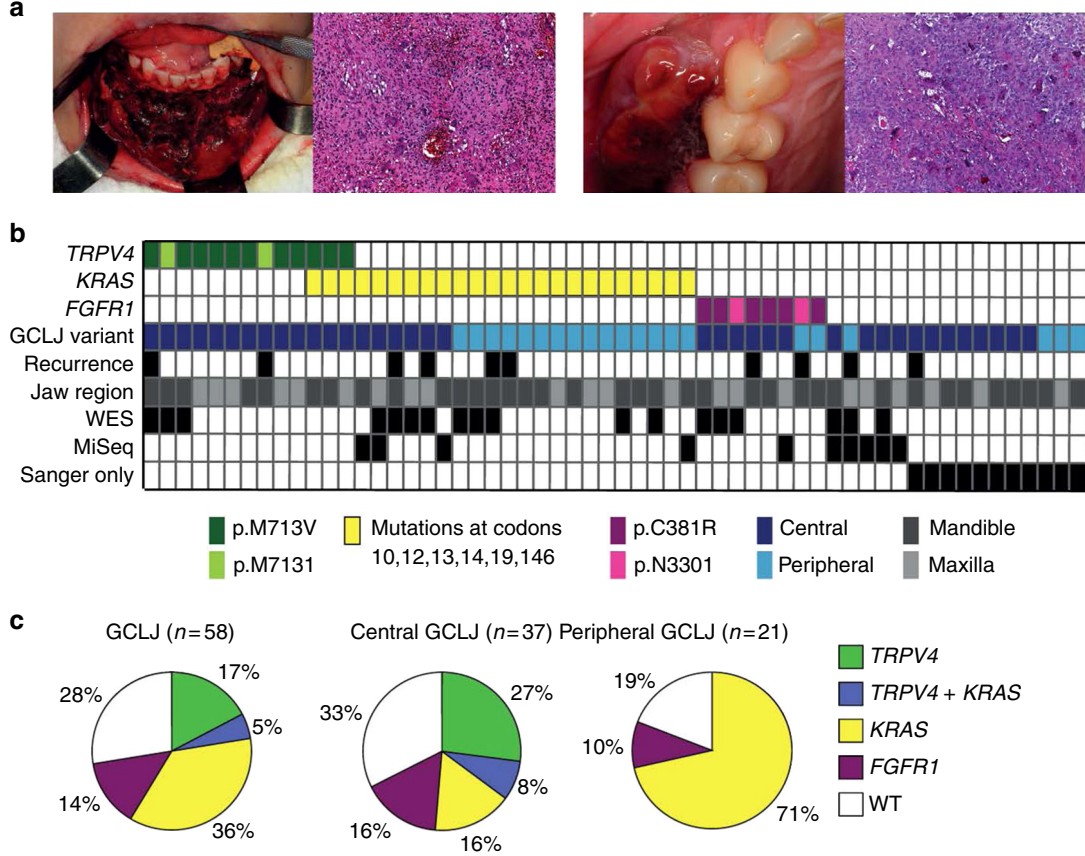

**Fig. 1** *TRPV4*, *KRAS* and *FGFR1* mutations drive central and peripheral giant cell lesions of the jaw (GCLJ). **a** Clinical image of an aggressive central GCLJ showing a large destructive bone lesion occurring in anterior mandible (left). Histologically, the lesion is composed of multinucleated osteoclast-like giant cells intermingled with oval to spindle-shaped mononuclear cells in a hemorrhagic stroma. Clinical image of a peripheral exophytic GCLJ (right). Histologically, the lesion is similar to that the central form. **b** Summary of *TRPV4*, *KRAS*, and *FGFR1* mutations identified in GCLJ. **c** Spectrum of *TRPV4*, *KRAS*, and *FGFR1* mutations in all GCLJ (left), central GCLJ, showing TRPV4 mutations are exclusive to this form (middle), and peripheral GCLJ (right)

ease of cycling between these states, thus altering ion channel activity. In the recently described low-resolution structure of TRPV4, M713 does not interact with other helices but rather faces the pore[18]. Based on this model, mutations at M713 are predicted to change the properties of the channel pore, and thus its function. However, the manner in which pore functioning would be altered is not readily predictable based on the current understanding of ion transport.

To confirm their impact on channel function, we stably overexpressed FLAG-tagged WT and p.M713V- or p.M713I-TRPV4 in HEK293 cells (Supplementary Fig. 4). Overexpression of mutant M713V- or M713I-TRPV4 markedly increased cell death compared to overexpression of WT-TRPV4, an effect that could be prevented by incubation with ion channel blocker ruthenium red (Fig. 2e). Next, we used patch-clamp electrophysiology and measured currents in response to voltage-ramps in cells exposed to physiological ionic conditions. Experiments were performed in the presence of ruthenium red to prevent calcium entry at negative voltages. Mutant TRPV4 cells showed drastically higher constitutive channel activity compared to WT-TRPV4-expressing cells. Indeed, basal currents increased by ~61% in M713V-TRPV4 and ~75% in M713I-TRPV4 compared to the current observed in WT-TRPV4 cells (Fig. 2f, g). Furthermore, in the presence of TRPV4 agonist GSK1016790a, significantly higher outward currents were recorded in p.M713V- and p.M713I-TRPV4 cells, with 41% and 64% respective increases compared to WT-TRPV4 (Fig. 2f, h). Use of the specific TRPV4 channel blocker GSK2193874 further confirmed that the recorded

outward current was due to TRPV4 channel opening (Supplementary Fig. 5). Collectively, these results indicate that p.M713V- and p.M713I-TRPV4 are gain-of-function mutations leading to increased channel activity.

**Somatic *KRAS* and *FGFR1* mutations are frequent in GCLJ.** Somatic heterozygous *KRAS* mutations were the most frequent, occurring in 41.3% (24/58) of GCLJ, predominantly in the peripheral form (15/21 compared to 9/37 in central) ($p = 0.0008$, Fisher's exact test) (Fig. 1b, c, Supplementary Data 1). Similar to TRPV4 alterations, these mutations have never been reported in GCLJ. *KRAS* mutations occurred mainly at known hotspots (p. G12D/A, p.G13D, p.A146V, p.A146P) and in five cases at rare alleles leading to p.V14L, p.L19F, or p.G10E (Supplementary Figs. 3, 6b). Notably, codon 146 (7/24), which is nearly selective for colorectal carcinomas in relation to other tumor types[19–21] and codon 12 (9/24), were the most frequently affected in GCLJ (Supplementary Fig. 6b). We also identified heterozygous somatic p.C381R and p.N330I gain-of-function mutations in *FGFR1* in eight cases (14%) (Fig. 1b, c; Supplementary Fig. 3). These mutations have previously been reported in osteoglophonic dysplasia[22,23] (OGD), a hereditary disease where GCLJ are seemingly absent. However, careful reading of published case reports[24] revealed that OGD patients with p.C381R and p.N330I mutations also presented with GCLJ, an association which had gone unnoticed. Importantly, none of our patients had clinical features suggestive of RASopathies, channelopathies or OGD, further confirming these mutations can be somatic in GCLJ.

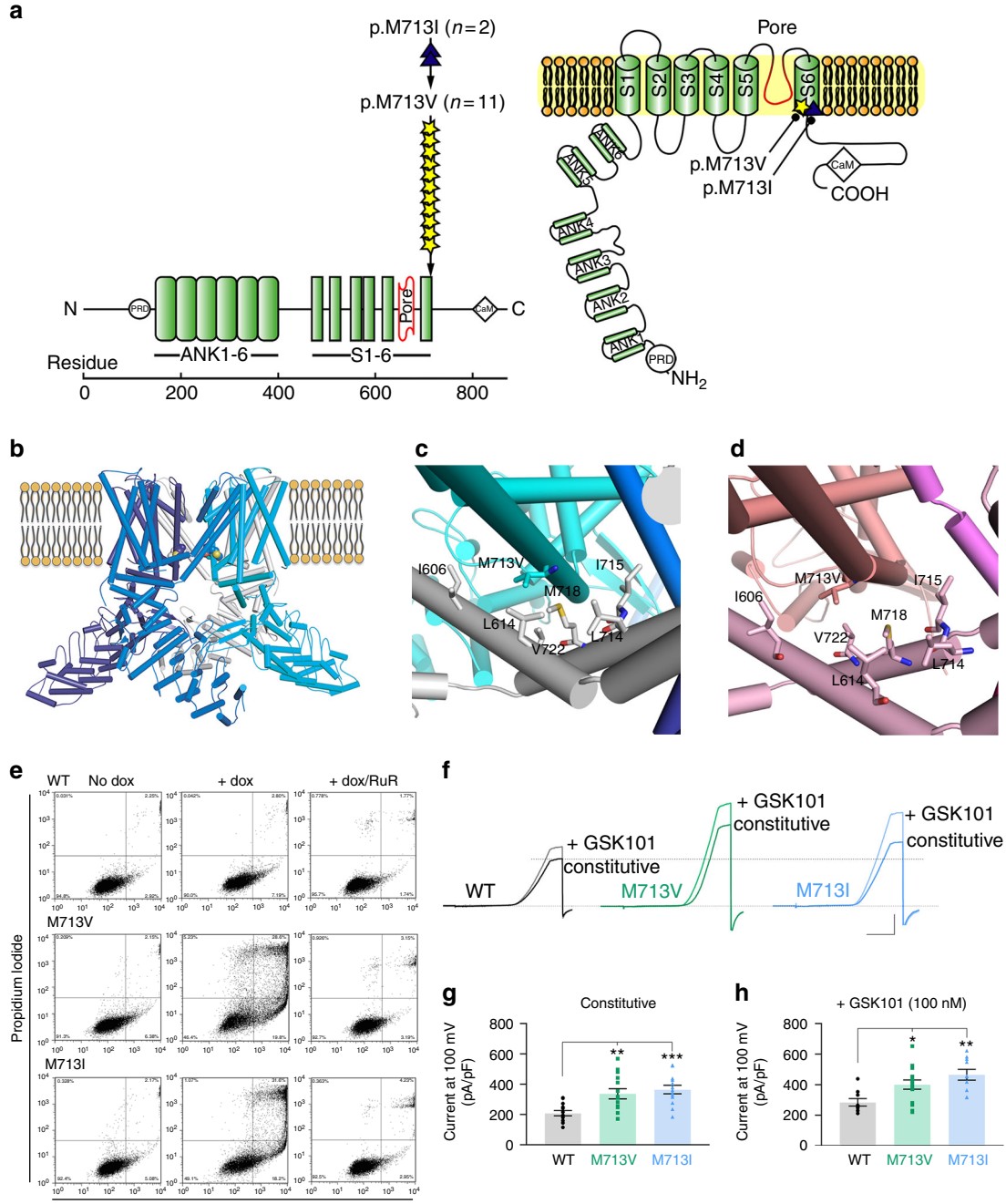

**Fig. 2** TRPV4 M713 mutations in GCLJ are predicted to affect channel function and are associated with increased channel activity. **a** Schematic diagrams of the TRPV4 channel protein domains, including six transmembrane segments (S1–6), pore-forming region, ankyrin repeat domains (ANK1–6), proline rich domain (PRD), and calmodulin (CaM)-binding site. The position of each TRPV4 mutation detected in GCLJ is represented by a star or a triangle, along with the number of affected cases. **b** Model of TRPV4 protein in its homo-tetrameric closed state with the sphere representation of M713 residue within the transmembrane domain. **c** Closed state of TRPV1 (PDB ID:3J5P) and **d** open state of TRPV1 (PDB ID:5IRX), modeled with TRPV4 M713V. Surrounding hydrophobic residues are shown; residues are labeled using TRPV4 numbering. **e** Cell death assay on HEK293 cells expressing exogenous wild-type (WT) and mutant (M713I and M713V) TRPV4. TRPV4 mutant proteins in HEK293 cells lead to increased cell death (middle), which could be prevented by incubation with the ion channel blocker RuR (right). The percentage of cells in each quadrant is indicated as follows: lower left, live cells; lower right, early apoptosis; upper right, late apoptosis; upper left, necrosis. Representative data of three biological replicates are shown. **f** Representative traces of TRPV4 currents recorded in HEK293 cells before (constitutive activity) and after the application of TRPV4 agonist GSK1016790a (GSK101, 100 nM). Currents were recorded using the conventional whole-cell configuration and 300-ms voltage ramps (−100 to 100 mV, from a holding potential of −50 mV); ruthenium red (RuR, 1 μM) was included in the bath solution. Vertical scale bar, 100 pA/pF; horizontal scale bar, 100 ms. **g** Individual-value plots of outward current recorded at 100 mV in the absence of GSK101 (mean ± s.e.m, $^{**}P < 0.01$, $^{***}P < 0.001$, one-way ANOVA followed by Dunnett's multiple comparisons test, WT, $n = 13$; M713V, $n = 14$; M713I, $n = 13$). **h** Individual-value plot of currents recorded at 100 mV from dialyzed HEK293 cells treated with 100 nM GSK101 and in the presence of 1 μM RuR (mean ± s.e.m, $^{*}P < 0.05$, $^{**}P < 0.01$, one-way ANOVA followed by Dunnett's multiple comparisons test, WT, $n = 9$; M713V, $n = 15$; M713I, $n = 10$). Black circles, WT; green squares, M713V; blue triangles, M713I

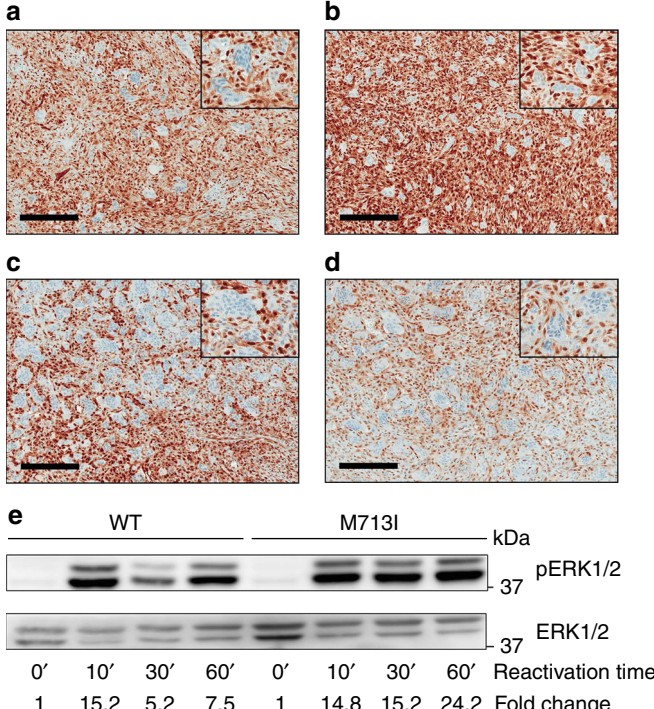

**Fig. 3** MAPK pathway activation in GCLJ. Immunohistochemical (IHC) staining for phospho-ERK1/2 shows strong positive staining in mononucleated cells in GCLJ lesions. Representative images of phospho-ERK1/2 staining in *TRPV4* p.M713V (**a**), *KRAS* p.G12D (**b**), and *FGFR1* p. C831R (**c**) mutant and in WT (**d**) GCLJ cases. **e** Immunoblot showing sustained phospho-ERK1/2 (pERK1/2) activation in *TRPV4* M713I HEK293 mutant cells compared to WT. A representative experiment of three independent assays is provided. Scale bar represents 200 μm

*FGFR1* mutations were mutually exclusive with *TRPV4* and *KRAS* mutations ($p = 0.0002$, Fisher's exact test, Supplementary Fig. 6a), while *TRPV4* or *KRAS* mutations co-occurred in 3 GCLJ samples and were mutually exclusive in 31 cases. Based on material availability for these three co-mutated samples, we could not perform further regional analysis and deep sequencing to confirm the co-existence of both mutations in tumor cells or identify a level of intratumor heterogeneity for these mutations.

**Driver mutations in GCLJ converge on MAPK activation**. *TRPV4*/Ca$^{2+}$ influx, *FGFR1* and *RAS* mutations are known to activate the MAPK pathway[25–29]. To assess MAPK activation in GCLJ samples, we examined phosphorylated ERK1/2 (phospho-ERK) immunoreactivity in samples with available material ($n =$ 34, Supplementary Data 1, Fig. 3a–d). Interestingly, all samples including triple negative GCLJ showed some level of positive staining for phospho-ERK1/2. Samples with either *TRPV4* (Fig. 3a), *KRAS* (Fig. 3b), or *FGFR1* (Fig. 3c) mutations had strong phospho-ERK staining in a large component of the mononuclear cells in the tumor, while the multinucleated giant cells were negative, as expected. A similar, albeit highly variable, pattern with significantly more patchy and lower phospho-ERK positivity in some samples was observed in GCLJ WT for these genes (triple negative, Fig. 3d). These findings indicate that MAPK pathway activation occurs in GCLJ including the tumors which carry the genetic alterations we identify in this entity.

To assess differential effects of mutant over WT TRPV4, we performed immunoblotting experiments and measured levels of phospho-ERK1/2 under serum starvation and subsequent time-course serum reactivation in HEK293 cells expressing WT-, p.

M713V-, or p.M713I-*TRPV4*. Sustained phospho-ERK1/2 activation was observed in p.M713I and p.M713V mutant cells compared to WT TRPV4, which showed decreased phospho-ERK1/2 activation after 30 min (Fig. 3e, Supplementary Fig. 6c). These data suggest that mutant TRPV4 leads to a more sustained activation of the MAPK pathway than the WT channel.

## Discussion

Overall, our work provides a genetic landscape for giant cell lesions of the jaw. We report somatic, heterozygous mutations in 72% of GCLJ in three genetic drivers: *TRPV4*, *KRAS*, and *FGFR1*. It is possible that our data underestimates the proportion of samples carrying a mutation in any of these 3 genes, as 11 samples were solely analyzed using standard sequencing, which may miss cases with lower mutational reads based on regional contamination with tumor microenvironment. Importantly, our findings reveal that despite histological similarities of GCLJ with GCTB, both entities are in fact genetically distinct, with distinct pathogenesis and activated pathways. While GCTB are characterized by a recurrent hotspot mutation in an epigenetic driver, G34W in histone 3.3, and high expression of RANKL[11,30], these are absent in GCLJ which harbor genetic alterations affecting signaling pathways, including MAPK pathway activation.

*TRPV4* is a broadly expressed polymodal Ca$^{2+}$-permeable channel. Germline mutations cause calcium entry malfunction and lead to hereditary channelopathies, a broad range of diseases affecting the skeletal or peripheral nervous systems[14,31–36]. These include skeletal dysplasias and diseases characterized by defects in bone development, osteonecrosis or arthropathies, and peripheral motor-sensory neuropathies, including Charcot–Marie–Tooth disease 2C. Notably, hereditary TRPV4 channelopathies are never associated with GCLJ and do not have mutations affecting the M713 residue which seem exclusive to this entity (Supplementary Fig. 2). *TRPV4* mutations seem exclusive to the central form of GCLJ, which interestingly occurs in the intramedullary component of the bone. Moreover, GCLJ are characterized by proliferation of mononuclear cells with osteoclast-type giant cells in a hemorrhagic vascular background. The known role of TRPV4 in promoting differentiation and inhibiting osteoclast apoptosis[37] and its role in modulating vascular function[38] are thus in keeping with a gain-of-function effect of *TRPV4* mutations and their role in GCLJ pathogenesis.

*KRAS* mutations are the most frequently identified genetic alteration in GCLJ. Codon 12 mutations, which are the most frequently observed in cancer[20], were also the most common *KRAS* mutations in our GCLJ cohort. Interestingly, a sizeable number of the less frequent mutation in codon 146, which is mainly found in colorectal cancer, was observed in peripheral GCLJ. Further analyses are required to assess if different *KRAS* mutant allele codon preferences shown by central (codon 12) and peripheral (codon 146) GCLJ are associated with different clinical behavior of these clinical variants. *FGFR1* mutations were mutually exclusive with *KRAS* and *TRPV4* mutations (Supplementary Fig. 6a). FGFR signaling is highly involved in bone growth and remodeling and several mutations affecting *FGFR1* have been implicated in cancer[39]. Interestingly, the somatic heterozygous p.C381R and p.N330I mutations we identify in GCLJ have been previously described in a germline disorder OGD[22,23], and never in sporadic tumors. Careful reading of OGD case reports indicate that individual with these mutations have GCLJ in keeping for a specific role of this genetic alteration in promoting these oral cavity tumors.

Collectively, our results demonstrate that *TRPV4*, *KRAS*, and *FGFR1* mutations converge on activating MAPK signaling in GCLJ. Germline activation of this pathway by any of these genes

(hereditary channelopathies, RASopathies, OGD) is invariably associated with skeletal alterations, in keeping with their potential role in the formation of GCLJ when they occur as somatic mutations in the oral cavity.

In summary, our results offer a genetic insight for targeted therapies in a maiming disease with currently limited therapeutic opportunities. Inhibitors targeting the TRPV4 channel are available with minimal side-effects in animal models and are being tested in clinical trials in diseases underlined by *TRPV4* alterations (NCT02497937, NCT03372603, and NCT02119260). In addition, FGFR and MEK inhibitors are being tested in several cancers[28]. We provide a solid pre-clinical frame for future clinical trials in this disease, and GCLJ patients could already benefit from available therapies targeting TRPV4, FGFR1 or the downstream activated MAPK pathway (e.g., MEK inhibitors) in recurrent and/or severely debilitating GCLJ.

## Methods

**Sample selection and characterization**. GCLJ were either collected fresh frozen in the Oral Medicine Clinic ($n = 6$) or as formalin-fixed paraffin-embedded (FFPE) tissue blocks from the Surgical Pathology files of the Faculty of Dentistry of Federal University of Minas Gerais (UFMG), Brazil ($n = 52$). The research was conducted in compliance with all relevant ethical regulations and the study was approved by the UFMG Ethics Committee and informed consent was obtained for cases collected prospectively. Information for all cases was acquired during clinical appointments or retrieved from patient files and included lesion size, pain complaint, teeth root resorption or displacement, cortical bone thinning, cortical bone perforation, and recurrence after curettage. All cases were sporadic lesions and the exclusion criteria included cherubism, hyperparathryroidism and any syndrome such as Noonan syndrome, Neurofibromatosis type 1 and Osteoglophonic Dysplasia. None of the patients presented common features of channelopathies such as short trunk, scoliosis or digital arthropathy, or motor and sensory neuropathies[14]. Plain radiographs, computed tomography, physical examination findings or clinical records were reviewed when available. All H&E slides were revised by two oral pathologists (R.S.G. and C.C.G.) to confirm diagnosis. The final diagnosis and classification into central or peripheral GCLJ was made on the basis of clinical and imaging examination combined with histopathological characterization. Central GCLJ were classified as aggressive or nonaggressive according to size, growth pattern, tooth resorption or displacement, cortical bone destruction or thinning, and recurrence[6–8]. Clinicopathological data for this cohort is presented in Supplementary Data 1 and Fig. 1a. For six cases collected prospectively (samples #1, 10, 13, 15, 35, and 36), a sample of peripheral blood, oral swab or normal oral mucosa was collected during surgery, to be used as germline DNA control. For cases #18 and #30, adjacent normal mucosa was used as germline DNA control.

**RNA-sequencing**. RNA from fresh tissue samples was extracted using AllPrep DNA/RNA Mini kit (Qiagen). RNA-Seq libraries were prepared from 1 μg of total RNA using the TruSeq Stranded Total RNA Sample Prep kit with Ribo-Zero Gold (Illumina). The quality and size of libraries was measured on an Agilent 2100 Bioanalyzer (Agilent Technologies). Libraries were then sequenced on an Illumina Hiseq 2000 platform to generate 100 bp paired-end reads. We used Trimmomatic[40] (v0.32) to remove adapter sequences, the first four bases of each read, and low-quality bases (phred33 < 30) at the end of each read. The reads were truncated once the average quality of a 4 bp sliding window fell below 30. An additional 3 bp were removed from both ends of each read if found to be of low quality. Short reads (<30 bp) produced as a result of trimming were discarded. The remaining clean set of reads were then aligned to the reference genome build hg19 (GRCh37) with STAR[41] (v2.3.0e) using the default parameters. Multimapping reads (MAPQ < 1) were discarded from downstream analyses.

**RNA-sequencing fusion calling**. Gene fusions were called using STAR-Fusion with default parameters. No fusions were detected in the six samples.

**RNA-sequencing variant calling**. RNA-seq variant calling was carried out as reported previously[42]. Reads spanning more than one exon were split using GATK's "split'N'Trim" functionality[43] (Genome Analysis Toolkit) (v3.2-2) and their mapping qualities, reassigned from 255 to 60. Indels were then realigned using GATK's "IndelRealigner". Variants were called using SAMtools mpileup[43,44] (v0.1.19) and annotated for hg19 refGene by Annovar[45] (February 2, 2016 version). Finally, to curb the incidence of false positives, we discarded calls that did not meet the following requirements: coverage >10 reads, alternative nucleotide count >3, SNV ratio >5%, indel ratio >15%, variant and mapping quality >15. The full list of variants detected is shown in Supplementary Data 3.

**Whole-exome DNA sequencing**. Fresh samples were cryosectioned, H&E stained slides of all FFPE and fresh tissue samples were revisited, and manual micro-dissection was carried out when needed before DNA isolation, to ensure lesion-rich cuts. Standard genomic DNA extraction was performed using commercial kits (Qiagen), following manufacturer's protocols. Exomes were captured using the Agilent SureSelect All Exon kit v5, according to the manufacturer's instructions. The enriched libraries were sequenced on the Illumina HiSeq 2500 with 125 bp paired-end reads. Sequence reads were mapped to the human reference genome (hg19) with Burrows–Wheeler Aligner[46], and duplicate reads were flagged using Picard (http://picard.sourceforge.net) and excluded from further analyses. Variants were called using three different variant callers including SAMtools mpileup[44], freebayes version v1.1.0-4-gb6041c6[47], and GATK haplotype caller[48] version 3.8 and were filtered to require at least 10% of reads supporting the variant call. To keep the high confidence variant calls, we only retained those variants that were called by at least two of three variant callers. Mutations were annotated using both ANNOVAR[45] and custom scripts. Annotated variants were filtered against the common germline polymorphisms present in dbSNP135, the 1000 Genomes project[49], NHLBI GO Exomes and inhouse database of approximately 3000 exomes previously sequenced. All variants were manually checked in Integrative Genomics Viewer (IGV). In addition, genes that are recurrently altered in our cohort, including *TRPV4*, *KRAS*, and *FGFR1*, were systematically inspected in WT samples using IGV to ensure that the bioinformatics pipeline did not miss any variants due to low mutant allele frequency. Tumor mutation burden (TMB) was calculated as reported previously.[50] Briefly, for each tumor/normal pair, variants specific to the tumor and not seen in the other normal samples of the set of pairs were filtered. To get tumor specific variants, we used raw read counts to remove all variants in the tumor where the matching normal has 3 or more reads supporting the variant. Then, we limited out list to (non)synonymous variants and short INDELs. Finally, we applied a 0.1% cutoff on population frequencies (ExAC, 1000 Genomes and EVS) and a 1% cutoff on our inhouse database of 3000 exomes. The somatic variants left were used to calculate the TMB by applying the following formula:

TMB/mb = # somatic variants/(30 MB × % of coverage [>10×]) assuming that the size of an exome is ~1% of the genome (Supplementary Data 2). WES basic statistics and full list of variants detected (excluding synonymous) are shown in Supplementary Datas 4 and 5, respectively.

**MiSeq targeted high-depth DNA sequencing**. We performed deep sequencing on six fresh-frozen sporadic central GCLJ samples using the Fluidigm Access Array and Illumina MiSeq system. The array system is based on an array-based PCR amplification of regions of interest. The panel covers exon 2 of H3.3 (3 H3.3 genes), coding regions of H3.1 and H3.2 isoforms (10 H3.1 and 3 H3.2 genes), as well as hotspot mutations in genes such as *IDH1* (codon 132), *IDH2* (codons 140 and 172), *ACVR1* (exons 6–9), *BRAF* (V600E), and *PPM1D* (exon 6). We achieved an average sequencing depth of over 8000×. In addition, those samples that showed low mutant allele peaks in chromatograms at *KRAS* exon 2 hotspots and *FGFR1* C381R were further validated by targeted amplicon sequencing. The target regions were amplified using primer pairs that were tagged with consensus sequences at the 5′ ends (CS1-F and CS2-R). The resulting amplicons were enriched and barcoded prior to sequencing on the MiSeq platform. The sequencing data were analyzed as previously described[51,52]. Eight cases WT for *TRPV4*, *KRAS*, and *FGFR1* mutations with available tissue were screened by MiSeq targeting *TRPV4* (codon M713), *FGFR1* (codons C381, N330), and *KRAS* (exon 2 and codon A146) to exclude the possibility of false negative due to low frequency of the mutant allele (Supplementary Fig. 1). Three out of these eight cases showed mutations (Supplementary Fig. 1, Supplementary Data 1), and five were confirmed WT. The remaining 11 WT cases where no tissue was available for MiSeq were Sanger sequenced only (Supplementary Fig. 1).

**Sanger sequencing**. All recurrent *TRPV4*, *KRAS*, and *FGFR1* mutations detected either by WES and or RNA-seq in the 19 samples screened by these methods were confirmed by Sanger sequencing. The 39 additional samples that were not sequenced by WES/RNA-seq were screened by Sanger sequencing to examine these mutations (Supplementary Fig. 1). Primers used to screen and validate *KRAS*, *TRPV4*, and *FGFR1* mutations were designed using Primer3 online software (http://bioinfo.ut.ee/primer3-0.4.0/). PCR products were bidirectionally sequenced on an ABI 3730XL DNA Analyzer (Applied Biosystems, USA) and resulting chromatograms were visualized using SnapGene software. Primers and PCR conditions are available upon request.

**In silico modeling**. TRPV4 mutations at residue M713 were modeled using the structure of TRPV1 from *Rattus norvegicus* (51% sequence identity). The corresponding residue for M713, M677 in TRPV1, was mutated to either isoleucine or valine. Mutations were modeled in both open state (PDB ID:5IRX) and closed state (PDB ID:3J5P) of the channel using PyMOL[16,17].

**Inducible overexpression of TRPV4 WT and mutant proteins**. A C-terminal Myc-DDK-tagged coding sequence of human TRPV4 (NM_021625) was purchased from Origene (RC220160) and cloned into the inducible pLVX-TetOne-Puro vector using in-fusion cloning (Clontech). TRPV4 M713I and M713V

(corresponding to c.2139G > A and c.2137A > G, respectively) mutants were generated by PCR based site-directed mutagenesis of the WT TRPV4 construct using the In-Fusion cloning kit. The constructs were transformed into Stellar competent cells, amplified and confirmed using restriction enzyme digestion (BamHI and EcoRI) and Sanger sequencing. To produce lentiviral particles, TRPV4 WT, M713I, or M713V expression constructs were co-transfected with packaging (pMDLg/PRRE, pRSV-Rev) and envelope (pMD2.g) protein plasmids into 293LTV cells. After three rounds of harvesting, lentiviral supernatant was pooled and filtered through 0.45-μm filters. Lentiviral particles were concentrated by ultracentrifugation, re-suspended in basal medium and stored at −80 °C. HEK293 cells were transduced with TRPV4 WT or mutant lentiviral particles and cells with stable expression were obtained by selection in 10 μg/ml puromycin.

**Cell culture**. HEK293 cells are routinely used for functional assays testing the effect of *TRPV4* mutation[34,35]. The HEK293 cell line was obtained from ATCC and GenePrint 10 System (Promega B9510) was used to authenticate the cell line. Cultured cells with stable TRPV4 WT or mutant expression were cultured in DMEM medium with 10% fetal bovine serum (FBS) and 1× penicillin/streptomycin in a 37 °C and 5% $CO_2$ incubator, in the presence of 10 μg/ml puromycin. HEK293 cells and 293LTV lentivirus producer cells were routinely tested and confirmed to be mycoplasma-free.

**Immunofluorescence and immunoblotting**. The expression of TRPV4 WT and p. M713I/V mutant proteins in HEK293 cells was induced with 0.1 μg/ml doxycycline (dox), in the presence of 10 μM ruthenium red to prevent death of TRPV4 mutant-expressing cells. Immunofluorescence and immunoblotting were performed 24 h after induction, following standard protocols. For immunofluorescence, the cells were incubated with anti-FLAG antibody (1:1000; CST 2368) overnight at 4 °C, followed by staining for donkey anti-rabbit AlexaFluor 594 secondary antibody (1:1000; ThermoFisher Scientific A-21207) for 1 h at room temperature. The cells were counterstained and mounted with ProLong Gold Antifade Mountant with DAPI (ThermoFisher Scientific). Images were acquired using a Zeiss LSM 780 laser scanning confocal microscope with a 63×/1.40 oil DIC objective. Immunoblotting was performed with the anti-FLAG tag (1:1000; CST 2368) and anti-GAPDH (1:50,000; Advanced Immunochemical 2-RGM2) antibodies at 4 °C overnight. Horseradish peroxidase-conjugated secondary antibody (1:5,000, GE healthcare NA934V) and the ECL detection kit (Amersham Biosciences) were used to detect immunoreactive material. For phospho-ERK1/2 assay, cells were serum starved (3% FBS) overnight, incubated for 90 min in the absence of RuR and FBS (0%), followed by serum reactivation (20% FBS) for 0, 10, 30, and 60 min. The membranes were incubated with the anti-phospho-p44/42 MAPK (1:500; phospho-Erk1/2, Thr202/Tyr204, CST 9101), anti-p44/42 MAPK (1:1,000; Erk1/2, CST 9102), and anti-beta tubulin (1:2,000; Abcam ab6046) at 4 °C overnight, followed by secondary antibody incubation and detection, as described above. The experiments were replicated three times. Original uncropped scans for all Western blots are shown in Supplementary Fig. 7.

**Electrophysiology**. Whole-cell currents were recorded in the conventional whole-cell configuration using a patch-clamp amplifier (Axopatch 200B; Molecular Devices), filtered at 1 kHz, digitized at 5 kHz, and stored on a computer for offline analysis with Clampfit 10.3 software. Whole-cell capacitance was measured with the cancellation circuitry in the voltage-clamp amplifier. Current density, obtained by dividing absolute current values (pA) by the capacitance (in pF), was used as a measure of activity. All electrophysiological recordings were performed at room temperature (~22 °C). Recording pipettes were fabricated by pulling (Narishige puller) borosilicate glass (1.5 mm outer diameter, 1.17 mm inner diameter; Sutter Instruments, USA). Pipettes were fire-polished to reach a tip resistance of ~4–6 MΩ. The bath solution consisted of: 134 mM NaCl, 6 mM KCl, 1 mM $MgCl_2$, 10 mM HEPES, 4 mM glucose, and 2 mM $CaCl_2$ (pH adjusted at 7.4). Pipettes were backfilled with a solution consisting of: 10 mM NaOH, 11.4 KOH, 128.6 mM KCl, 1.09 mM $MgCl_2$, 2.2 mM $CaCl_2$, 5 mM EGTA, and 10 mM HEPES (pH adjusted at 7.2). Currents were recorded before (constitutive) and after the application of the TRPV4 channel agonist GSK1016790a (GSK101, 100 nM). Ruthenium red (RuR,1 μM) was included in the bath solution to block TRPV4-mediated $Ca^{2+}$ influx and prevent $Ca^{2+}$ overload without affecting outward currents. RuR-mediated block is voltage-dependent and is reversed at depolarized membrane potentials thus allowing the monitoring of outward TRPV4 currents at 100 mV.

**Cell death assay**. Twenty-fours hour prior to the assay, cell culture media was replaced as follows: no doxycycline (dox)/no Ruthenium Red (RuR); 0.1 μg/ml dox; 0.1 μg/ml dox + 10 μM RuR. Apoptosis was detected using a commercially available kit (556547, BD Biosciences), following the manufacturer's recommendations. Fluorescence from FITC-annexin V and propidium iodide was measured by flow cytometry using a FACSCalibur FL-1 and FL-3 channels, respectively, and the CellQuest Pro software. Statistical quadrant analysis was done using FlowJo 10.4 software and normalized with 10,000 live cells. The experiment was performed three times.

**Immunohistochemistry**. FFPE samples from 34 GCLJ patients with available material were immunostained for phospho-ERK1/2 (Supplementary Data 1). The Discovery XT Autostainer (Ventana Medical System) was used and heat-induced epitope retrieval with CC1 prediluted solution (Ref. 950-124) was performed in the slides after de-paraffinization, following standard protocols. All solutions used for automated immunohistochemistry were from Ventana Medical System (Roche) unless otherwise specified. Slides were incubated with rabbit monoclonal anti-pERK1/2 (1:100, Thr202/Tyr204, CST 4376) for 32 min at 37 °C. Negative control was performed by the omission of the primary antibody. Slides were counterstained with hematoxylin, dehydrated through graded alcohols, cleared in xylene, and mounted with mounting medium (Eukitt, Fluka Analytical). Sections were scanned at 40X using the Aperio AT Turbo Scanner (Leica Biosystems). Nuclear and cytoplasmic staining were considered a positive reaction. The extent of staining in the tumor cells was evaluated by an oral pathologist (R.S.G.) and >10% of tumor positive cells was considered as positive immunostaining.

## Data availability
Primary WES and RNA-Seq data has been deposited to the European Genome-phenome Archive (EGA) at https://www.ebi.ac.uk/ega/home with Accession no. EGAS00001002910. All relevant data are available from the authors upon reasonable request.

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

## Acknowledgments

The authors thank Grazielle Menezes and Aline Cruz for their assistance with the acquisition of patient information. C.C.G. (Proc. 88881.118879/2016-01) and R.S.G. (Proc. 88881.119257/2016-01) are research fellows at Coordination for the Improvement of Higher Education Personnel (CAPES)/Brazil. E.T.V. (Proc.2015/20142-0) is research fellow at FAPESP (Fundação de Amparo a Pesquisa do Estado de São Paulo). T.G. is a recipient of CIHR postdoctoral fellowship. This work was supported in part by funding from the Brazilian National Council for Scientific and Technological Development (C.C. G. and R.S.G.); NIH (P01CA196539 to N.J. and J.M.); the Canadian Institutes of Health Research (CIHR MOP286756 to N.J.); ICHANGE consortium, and the Fonds de Recherche du Québec-Santé (salary awards to A.B., J.P., C.L.K., N.D.J., and N.J.). This work was also supported by a postdoctoral fellowship (17POST33650030 to O.F.H.) from the American Heart Association, the Totman Medical Research Trust (to M.T.N.), Fondation Leducq (to M.T.N.), European Union's Horizon 2020 research and innovation program (Grant agreement no. 666881, SVDs@target, to M.T.N.), and NIH (P01-HL-095488, R01-HL-121706, R37-DK-053832, 7UM-HL-1207704, and R01-HL-131181 to M.T.N.).

## Author contributions

C.C.G., T.G., A.B., O.F.H., J.P., L.G.M., E.T.V., M.G.D., B.R., and H.H. performed experiments. H.N., J.M., N.D.J., P.S.O., D.S., C.L.K., A.V.B., A.M.B., and E.B. performed bioinformatic analyses. C.C.G., T.G., A.B., O.F.H., M.G.D., M.T.N., J.M., R.S.G., and N.J. performed data analyses and generated the text and figures. C.C.G., T.G., M.G.D., W.H. C., R.S.G., and N.J. collected data and provided patient materials. C.C.G. drafted the manuscript. J.M., R.S.G., and N.J. provided leadership for the project. All authors contributed to the final manuscript.

## Additional information

**Competing interests:** The authors declare no competing interests.

