## [Peer Review File · Nature Communications]

Reviewers' Comments:

Reviewer #1:

Remarks to the Author:

The authors have studied 58 cases of giant cell lesion of the jaw (GCLJ). Though "benign" they are significant problems for those affected often with major functional consequences. This study identifies driver mutations in TRPV4, KRAS and FGFR1. TRPV4 is of particular interest as mutations of this gene are known to cause a number of other bone diseases. The functional consequences of the TRPV4 mutations have been characterized using electrophysiological methods. The data is largely convincing and this study represents an important step forward in the investigation of GCLJ. The authors should address the following points.

1) A significant subset (19/58) of samples have no mutations in any of the three drivers identified. Of these, only 3 had exome sequence. That suggests a bias towards false negative results in the validation sequencing. There is no clear description of the method used to "validate and extend..." (bottom of p3.) which links that statement to the methods. It seems likely that the validation method used (capillary sequencing?) was of inadequate sensitivity for the dilution levels of some samples. Was sample purity assessed? It is important to distinguish technical issues from true negatives to enhance the value of this study for others interested in these tumors. This issue would also affect the potential enthusiasm for and impact of targeted therapy. On the other hand, if nearly a third of cases really lack an identified driver, then future studies would be important to study these "triple negative" tumors.

2) In the supplement, list the samples which were validated for KRAS and FGFR1 by Fluidigm/Miseq deep sequencing and their VAFs for those genes. TRPV4 was not listed as being validated in that assay. Is that correct? If so, it should be clearly so stated.

3) Basic statistics for the WES, particularly read depth should be added.

4) Any other mutations of interest found by WES outside of the 3 featured genes should be listed in a supplemental Table.

5) Table S1. VAF should be added for each sample where it is available.

6) There is no mention of how the primary sequence data will be made publicly available.

Reviewer #2:

Remarks to the Author:

Remarks to the Editor:

In this manuscript by Gomes et al that authors sequence giant cell lesions of the jaw (GCLJ) by targeted sequencing, whole exome sequencing and RNAseq. They discover driver mutations in KRAS and FGFR1 and a putative driver mutation in TRPV4 (M713I/V). The work is novel and represents an important resource for researchers. Additionally, the study identifies two potential druggable driver mutations in FGFR1 and TRPV4, thus there is clear clinical implications for the work. However, this is a very short report and could benefit from additional experimentation to characterize the TRPV4 mutant more rigorously and in general provide additional sequencing details for all the GCLJ samples analyzed.

Comments:

1. The allele frequencies of all the mutants are very low? Is this due to a high percent of normal cells being present in the tumor samples? It appears every single mutation in KRAS, FGFR1, and

TRPV4 is a heterozygous mutation - the authors should address why these mutations are never homozygous. The chromatograms displaying the mutations in Figure S2 also confirm extremely low allele frequencies for the mutants. Much lower than driver mutations for other types of cancer. What is the spectrum of mutations and the mutational burden in GCLJ?

2. Similarly, the study would benefit from an analysis of the tumor heterogeneity, are the TRPV4 and KRAS mutations occurring in the same cancer cells? Are the FGFR1, KRAS, and TRPV4 mutations truncal driver mutations or branch mutations?

3. Figure 1D does not add much value to the study and the authors should attempt molecular dynamics simulations of the M713V/I mutants to determine how the mutation effects the movement of nearby amino acids in both the open and closed states of the channel as this information would be very informative.

4. Why does the mutant induce apoptosis in the 293T cells and the WT does not affect cell death (somewhat counterintuitive)? The authors should consider performing these experiments on a more relevant cell type, such as osteoclasts, where TRPV4 expression inhibits apoptosis - would this protective effect on osteoclasts be even greater in cells overexpressing the mutants? Is it possible to isolate GCLJ cells from patients and assess the effects of the TRPV4 inhibitor or FGFR1 inhibitors? Alternatively knocking down the mutants and assessing the effects on GCLJ cell survival and proliferation could also be informative.

5. All mutations identified by targeted sequencing, RNAseq, and WES should be included as a supplemental table.

6. What is the mechanism by which mutant TRPV4 is promoting formation of these giant cell lesions. An analysis of the signaling pathways activated downstream of WT and mutant TRPV4 would benefit the paper tremendously. Preferably if this could be assessed in cell lines derived from patient samples through use of TRPV4 inhibitor or knock-down or in osteoclasts overexpressing the WT or mutant TRPV4.

Reviewers' comments:

Reviewer #1 (Remarks to the Author):

The authors have studied 58 cases of giant cell lesion of the jaw (GCLJ). Though “benign” they are significant problems for those affected often with major functional consequences. This study identifies driver mutations in TRPV4, KRAS and FGFR1. TRPV4 is of particular interest as mutations of this gene are known to cause a number of other bone diseases. The functional consequences of the TRPV4 mutations have been characterized using electrophysiological methods. The data is largely convincing and this study represents an important step forward in the investigation of GCLJ.

We would like to thank the reviewer for his interest in our manuscript and for highlighting the importance and relevance of our findings.

The authors should address the following points.

1) A significant subset (19/58) of samples have no mutations in any of the three drivers identified. Of these, only 3 had exome sequence. That suggests a bias towards false negative results in the validation sequencing. There is no clear description of the method used to “validate and extend...” (bottom of p3.) which links that statement to the methods. It seems likely that the validation method used (capillary sequencing?) was of inadequate sensitivity for the dilution levels of some samples. Was sample purity assessed? It is important to distinguish technical issues from true negatives to enhance the value of this study for others interested in these tumors. This issue would also affect the potential enthusiasm for and impact of targeted therapy. On the other hand, if nearly a third of cases really lack an identified driver, then future studies would be important to study these “triple negative” tumors.

The reviewer raises very valid points. We agree that we cannot exclude false negatives in the initial 19 samples tested wild-type using Sanger sequencing in our series. Indeed, Sanger can only detect mutations when the mutational burden is $\sim >20\%$, a threshold difficult to achieve in cases heavily contaminated with normal infiltrating cells from the microenvironment. To this effect, all samples included in this study had been manually microdissected, either matched with an H&E slide (FFPE cases) or frozen section control (fresh tumor samples) to ensure tumor enrichment prior to DNA extraction. Two of the co-authors (C.C.G. and R.S.G.) are oral pathologists and carefully revised all samples and tissue blocks to guarantee nucleic acid extraction was carried out in representative tumor areas where they strove to have more than 50% tumor material in a given case. Regardless, there is always a level of normal tissue contamination as these lesions are characterized by numerous hallmark giant cells which are reactive to the intermingled mononuclear cells, the proliferative component of the tumor and the rich cellular fraction at the basis of the disease, along with endothelial cells, macrophages and other immune cells (please also see new Figure 3 a-d, which illustrates this pathology). This makes the possibility of having false negatives in the absence of deep-sequencing quite real (please see also below). Importantly, in the closely related giant cell lesions of the bone, the driver histone 3.3 G34W mutation is also absent from the giant cell component and only present in a subset of stromal cells (Behjati *et al.*, *Nature Genetics* 2013, our reference 7). Next Generation Sequencing showed that in some samples, this histone mutation had low variant allele frequency (VAF $\sim 10-20\%$), based on its dilution with the tumor microenvironment. This is still the genetic event responsible for the disease and the reactive stroma, and these findings mirror what we also see for the mutations we identify in some samples.

To limit false negatives, we were able to acquire quality material for 4 of 15 cases tested only by Sanger and repeated the analysis using deep-sequencing (MiSeq on all 4 testing for TRPV4 (M713),

***KRAS* (exon 2 and A146) and *FGFR1* (C381 and N330)). One case carried *FGFR1* p.C381R, another *KRAS* p.G10E and another *KRAS* p.A146P, while the last was wild-type for all these genes. Thus, there is a proportion of samples that is indeed potentially wild-type for *TRPV4*, *FGFR1* and *KRAS*, as five of the original 19 wild-type cases are triple negative when assessed by deep-sequencing (WES and MiSeq, Figure 1b/Supplementary Figure 1/Table 1). We also inspected these genes in each BAM file in the 3 WT samples originally screened by WES to investigate whether the pipeline missed the mutations, but all were real “triple negative”. We cannot exclude that a portion of the 11 remaining wild-type samples assessed by Sanger sequencing only based on material availability carry in fact one of these mutations. To be clear to the readers, in the revised version of the manuscript, we have included a flowchart (Supplementary Figure 1) on sequencing approaches used on the study samples, and in Figure 1b, we introduced lanes showing which type of analyses were performed. The possibility of false triple negatives was also made clear in the discussion of our revised manuscript.**

2) *In the supplement, list the samples which were validated for KRAS and FGFR1 by Fluidigm/Miseq deep sequencing and their VAFs for those genes. TRPV4 was not listed as being validated in that assay. Is that correct? If so, it should be clearly so stated.*

We thank the reviewer for this suggestion. We have included all assays used for mutation detection or validation for each sample in Supplementary Table 1 of the revised manuscript. We included a flowchart with this information in supplementary materials (new Supplementary Figure 1) to guide the reader. *TRPV4*, *KRAS* and *FGFR1* mutations detected by NGS in the 19 samples screened by these methods were confirmed by Sanger sequencing and when possible also by MiSeq.

3) *Basic statistics for the WES, particularly read depth should be added.*

As suggested by the reviewer, we have now included this information in a Supplementary Table 4.

4) *Any other mutations of interest found by WES outside of the 3 featured genes should be listed in a supplemental Table.*

As requested by the reviewer, Supplementary Table 5 listing all variants detected by WES is now included in the revised version of the manuscript. In addition, we also included a Supplementary Table 3 with the variants detected by RNA-seq.

5) *Table S1. VAF should be added for each sample where it is available.*

We have included VAF for all samples whenever this information was available.

6) *There is no mention of how the primary sequence data will be made publicly available.*

Primary sequencing data has been deposited at the European Genome-phenome Archive (EGA) at <https://www.ebi.ac.uk/ega/home> with accession number EGAS00001002910.”

Reviewer #2 (Remarks to the Author):

Remarks to the Editor:

In this manuscript by Gomes et al that authors sequence giant cell lesions of the jaw (GCLJ) by targeted sequencing, whole exome sequencing and RNAseq. They discover driver mutations in *KRAS* and *FGFR1*

and a putative driver mutation in TRPV4 (M713I/V). The work is novel and represents an important resource for researchers. Additionally, the study identifies two potential druggable driver mutations in FGFR1 and TRPV4, thus there is clear clinical implications for the work. However, this is a very short report and could benefit from additional experimentation to characterize the TRPV4 mutant more rigorously and in general provide additional sequencing details for all the GCLJ samples analyzed.

We would like to thank the reviewer for the positive comments and the valuable suggestions. We have incorporated new data in the revised manuscript, in addition to more sequencing details. As also suggested, we have additionally expanded the manuscript to include more experimental details and new functional data.

Comments:

1. The allele frequencies of all the mutants are very low? Is this due to a high percent of normal cells being present in the tumor samples? It appears every single mutation in KRAS, FGFR1, and TRPV4 is a heterozygous mutation - the authors should address why these mutations are never homozygous. The chromatograms displaying the mutations in Figure S2 also confirm extremely low allele frequencies for the mutants. Much lower than driver mutations for other types of cancer. What is the spectrum of mutations and the mutational burden in GCLJ?

We would like to thank the reviewer for his comment that is similar to the first comment raised by reviewer 1 (please also refer to our answer to this question above). Tumor mutation burden (TMB) was calculated as reported previously (Chalmers et al., *Genome Medicine* 2017, 9:34 <https://doi.org/10.1186/s13073-017-0424-2>) for the 5 cases for which we had WES data for tumor and matched germline DNA. Consistent with the benign nature of these lesions, TMB was ~1 per MB in the 5 cases, a similar low TMB to the giant tumor lesions of the bone previously analyzed by another group (Behjati et al. *Nature Genetics* 2013, our reference number 7). This information is now shown in Supplementary Table 2.

These mutations are somatic and heterozygous with variable allele frequency that ranges between 10% for a few samples to up to 64% (Supplementary Table 1). This heterozygous, somatic status of the mutations we identify mirrors the histone 3 mutation (G3W) detected in the closely related giant cell tumors of the bone (GCTB) (Behjati et al, *Nature Genetics*, 2013, our reference 7). All cases reported in this study were from sporadic lesions, as we excluded cases from syndromic patients. In addition, even though these tumors are benign, on histopathological examination, they exhibit some level of infiltration by the tumor microenvironment composed of endothelial cells, fibroblasts and occasional inflammatory cells which results in varying degrees of intratumor heterogeneity. Although we selected representative areas enriched for apparent tumor cells, the level of contamination by the microenvironment could affect the frequency of mutations in our cohort. Our findings are in line with the low VAF of *H3F3A* mutations in tumor cells of GCTB and chondroblastoma (Behjati et al, *Nature Genetics*, 2013, our reference 7), which are histologic mimics of GCTJ. *FGFR1* and *KRAS* mutations in rare germline syndromes are always heterozygous and represent gain-of-function mutations, and this is also true for *TRPV4* mutations in hereditary channelopathies. This observation is in keeping with the data we obtained in giant cell lesions of the jaw.

In the Discussion section of our revised manuscript, we have included a paragraph addressing these possibilities and now provide a flowchart (new Supplementary Figure 1), a revised Figure 1, Supplementary Table 1 and have expanded on these limitations and possible explanations of the lower VAFs of some tumors.

2. Similarly, the study would benefit from an analysis of the tumor heterogeneity, are the TRPV4 and

KRAS mutations occurring in the same cancer cells? Are the FGFR1, KRAS, and TRPV4 mutations truncal driver mutations or branch mutations?

We agree with the reviewer that this is an important point to address. As noted above, there is already a degree of intra-tumoral heterogeneity based on the level of infiltration with the tumor microenvironment. We identify 3 cases carrying both *TRPV4* and *KRAS* mutations, even though these mutations seemed to occur in a mutually exclusive pattern in the clear majority of the other cases (Supplementary Figure 6a). This is a retrospective cohort and unfortunately, we did not have sufficient quality material from any of these samples to do regional investigation of the mutations in different tumor areas to infer clonality and/or intra-tumor heterogeneity. We have nonetheless amended the discussion in our revised manuscript where we suggest that further studies are warranted to assess the co-existence of both mutations in the same cell or their individual presence in tumor cells of distinct lineage or geographic areas.

3. Figure 1D does not add much value to the study and the authors should attempt molecular dynamics simulations of the M713V/I mutants to determine how the mutation effects the movement of nearby amino acids in both the open and closed states of the channel as this information would be very informative.

We agree with the reviewer that molecular dynamics simulations could potentially provide additional mechanistic insights in how the mutants impact the channel properties. However, molecular dynamics simulations of the open and closed states, as suggested, will unlikely be informative. Given the very conservative nature of the mutations, it is extremely doubtful that movement of nearby amino acids will be significantly impacted. One molecular dynamics analysis that may prove useful is “end-point targeted molecular dynamics” where the process of opening and closing of the channel is simulated. This approach has indeed successfully been used to examine various channels and the impact of mutations (e.g. PMC3642040, PMC3896152, PMC5430913). But these examples also show that such an analysis is far more involved than what is proposed by the reviewer and would fall outside the scope of this manuscript. Moreover, given the discrepancies between the two structural models available for TRPV4, it is arguably premature to initiate such an extensive analysis at this point.

4. Why does the mutant induce apoptosis in the 293T cells and the WT does not affect cell death (somewhat counterintuitive)? The authors should consider performing these experiments on a more relevant cell type, such as osteoclasts, where TRPV4 expression inhibits apoptosis - would this protective effect on osteoclasts be even greater in cells overexpressing the mutants? Is it possible to isolate GCLJ cells from patients and assess the effects of the TRPV4 inhibitor or FGFR1 inhibitors? Alternatively knocking down the mutants and assessing the effects on GCLJ cell survival and proliferation could also be informative.

Our results are in line with results from previous studies, which showed that expression of *TRPV4* gain-of-function mutations in HEK293 cells leads to increased cell death due to increased intracellular calcium levels (Landoure *et al.*, *Nature Genetics* 2010;42:170-174). Electrophysiology showed that both M713 mutants exhibited increased channel activity, compared to the WT, and this was consistent with protein modeling predictions. Although cell death was markedly increased in the cells expressing mutant M713V- or M713I-TRPV4, apoptosis was also induced in WT-TRPV4 expressing cells, but to a lesser extent. This is consistent with the differences in channel activity observed in the patch-clamp experiments. Also, HEK293 cells have been consistently used in several papers assessing functional effect of TRPV4 mutation (Landoure *et al.*, *Nature Genetics* 2010;42:170-174; Rock *et al.*, *Nature Genetics* 2008;40:999-1003; Mah *et al.*, *J Med Genet.* 2016;53:705-709; Lamande *et al.*, *Nature Genetics* 2011;43:1142-1146).

Respectfully, we think that using osteoclasts to test the effects of TRPV4 could be misleading to the readers, as it would suggest that giant cells are the main cell type of this lesion. Although giant cell lesions of the jaw show osteoclast phenotype, the main parenchymal cells of this lesion are the mononuclear cells, often referred to as “stromal” cells.

5. All mutations identified by targeted sequencing, RNAseq, and WES should be included as a supplemental table.

As requested by both reviewers, we have included in our revised submission a full list of all variants identified in the RNAseq and WES, as well as those identified by targeted sequencing. These results are now in our revised Supplementary Tables 1, 3 and 5.

6. What is the mechanism by which mutant TRPV4 is promoting formation of these giant cell lesions. An analysis of the signaling pathways activated downstream of WT and mutant TRPV4 would benefit the paper tremendously. Preferably if this could be assessed in cell lines derived from patient samples through use of TRPV4 inhibitor or knock-down or in osteoclasts overexpressing the WT or mutant TRPV4.

Calcium influx and TRPV4 activate MAPK-ERK pathway (Rosen *et al.*, *Neuron* 1994;12:1207-1221; Chen Y *et al.*, *Pain* 2013;154:1295–1304; Chen Y *et al.*, *J Biol Chem* 2016;6:291:10252-10262), similar to KRAS and FGFR activation which also lead to increased MAPK-ERK signaling. To this effect, we assessed phosphorylation of downstream ERK1/2 as a surrogate marker of the MAPK pathway activation in tumor samples and the HEK293 cell lines we had manipulated to express mutant or wild type TRPV4. Immunohistochemical staining for phospho-ERK1/2 in the 34 giant cell lesions of the jaw where we had available slides showed that all displayed some level of positive p-ERK1/2 immunostaining in the tumor mononuclear cells, especially samples carrying a given mutation (New Figure 3a-d). These results indicate that the reported mutations lead to activation of the MAPK pathway and that samples apparently wild-type (possible false negative for some, see above) could have alterations to be uncovered in follow-up cohorts that lead to activation of this pathway.

HEK293 cells overexpressing M713I and M713V TRPV4 showed a strong and sustained phospho-ERK1/2 activation using western blot analyses. Conversely, HEK293 overexpressing the wild-type TRPV4 showed initial strong phospho-ERK1/2 activation which, as expected, decreased after 30 min. This further supports our observation that the MAPK pathway is activated in cells by all three mutations and could be therapeutically targeted. These results are included in our new Figure 3e, Results and Discussion sections as well as in the supplementary material of the revised manuscript we are submitting (Supplementary Figure 6c).

We would like to thank the reviewers for their constructive comments that we hope to have satisfactorily addressed.

Reviewers' Comments:

Reviewer #1:

Remarks to the Author:

The authors have significantly improved the manuscript in response to the previous critique.

Reviewer #2:

Remarks to the Author:

The authors have adequately addressed all of my concerns and I now recommend this manuscript for publication.